

# Dynamic perspectives on biodiversity quantification: beyond conventional metrics

Manjula Josephine Bollarapu[1], Swarna Kuchibhotla[1], Ramarao Kvsn[2] and Harshita Patel[3]

[1] Department of Computer Science and Engineering, Koneru Lakshmaiah Education Foundation, Vaddeswaram, Guntur, Andhra Pradesh, India
[2] Department of Computer Science and Engineering, Chandigarh University (Deemed To be University), Chandigarh, Punjab, India
[3] School of Computer Science Engineering and Information Systems, Vellore Institute of Technology, Vellore, Tamil Nadu, India

## ABSTRACT

Our research addresses the pressing need to assess biodiversity in the face of increasing habitat destruction and species extinctions. Several researchers have modelled conventional measures to assess biodiversity. Every measure evaluates biodiversity by considering different properties. Among them Simpson and Shannon indices are widely used, they primarily focus on species richness and abundance, overlooking the importance of rare or unique species. This limitation makes it challenging to identify which species drive changes in biodiversity and hampers conservation efforts. Moreover, these measures are sensitive to sample size and biased towards dominant species, leading to inaccurate estimations. To overcome these challenges, we propose a novel mathematical model that provides a comprehensive assessment of biodiversity. Our model accounts for species dominance, addresses sample size sensitivity, and highlights the significance of rare species within a community. By applying our measure to real-time scenarios and comparing it with traditional methods using the same dataset, proposed measure demonstrated its efficacy in capturing biodiversity dynamics over time.

## INTRODUCTION

Biodiversity encompasses the entire spectrum of life forms on Earth, with each species playing a vital role in sustaining healthy ecosystems. It encompasses all forms of life, and each of the species that exists in this biodiversity plays a crucial role in maintaining a healthy ecosystem. But in recent times, because of human alterations of the global environment we can observe the extinction of many species which is leading to the decline of biodiversity (*Gadgil et al., 1993*). This decline in biodiversity will alter ecosystem processes and bring many environmental changes that will have a great impact on the services that humans require to derive from the ecosystem for their sustainability. Therefore, maintaining healthy biodiversity is essential for the sustainability and well-being of all living species

Corresponding author
Harshita Patel,
harshita.patel@vit.ac.in

on Earth. Because of this day-to-day decline in biodiversity, rapid ecological changes are taking place which paved the way for the quantitative measurement of biodiversity for its future conservation (*Chakravarthi, Reddy & Ramaiah, 2017*; *Dawson et al., 2011*). To assess biodiversity, many great ecologist researchers have proposed various mathematical measures. Each measure evaluates biodiversity by considering different properties (*Heydari, Omidipour & Greenlee, 2020*).

However, several existing measures are static in nature because they do not assess biodiversity on a time-series basis. This static nature of the measures fails to provide insights into actual changes in biodiversity. Hence, to overcome this flaw and to understand the changes in the biodiversity dynamically, there is a strong need to develop a robust measure that assess biodiversity periodically (*Pollock et al., 2020*). In line with this, this article proposes a new mathematical model to assess biodiversity over time, addressing the issues with static measures.

The structure of this article unfolds as follows: Section 'Related Work' outlines traditional measures that are commonly used to evaluate species biodiversity. Section 'Methodology' delineates the systematic development process employed to devise the proposed measure. Section 'Exploring real-time use case scenarios-synthetic data modelling' provides a comprehensive overview of various real-time scenarios, their corresponding use-cases, and the synthetic data utilized for assessing biodiversity across different time periods. In Section 'Unveiling measurement gaps: Simpson & Shannon measures', traditional measures are applied to the synthetic data, revealing inherent limitations. Section 'Exploring Proposed Measure' delves into a detailed discussion of the proposed measure. Section 'Comparative analysis: proposed measure versus traditional methods' conducts a comparative analysis between the proposed measure and traditional methods, followed by a thorough examination of the results obtained.

## RELATED WORK

As previously mentioned, numerous researchers have put forth various mathematical measures (*Beaugrand, Kirby & Goberville, 2020*) for evaluating biodiversity as shown in Fig. 1. This study aims to comprehensively analyze these existing mathematical measures used for assessing biodiversity and provides a detailed exploration of their practical applicability in real-world scenarios.

### Simpson's index
Simpson's index (*Simpson, 1949*), introduced by Edward Hugh Simpson in 1949, is a biodiversity assessment method that incorporates both species abundance and richness when evaluating biodiversity.

$$D = \sum n(n-1)/N(N-1). \tag{1}$$

### Shannon's diversity index
The Shannon measure (*Shannon & Weaver, 1949*), introduced by Claude Shannon and Warren Weaver in 1949, is another widely utilized method for assessing biodiversity. Like
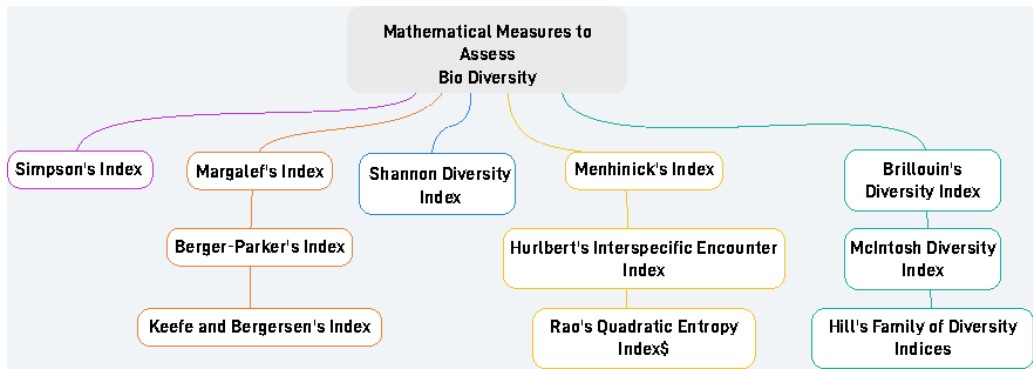

**Figure 1** Various mathematical measures to assess biodiversity.

Simpson's index, Shannon's diversity considers both species richness and evenness when assessing biodiversity.

$$H = -\sum_{i=1}^{s} PilnPi. \tag{2}$$

## Brillouin's diversity index

The Brillouin index, developed by L. Brillouin in 1962 (*Brillouin, 2013*), is a biodiversity measure that primarily focuses on species richness. It is defined using the formula below.

$$H_B = ln(N!) - \sum ln(n_i!)/N. \tag{3}$$

## Margalef's diversity index

Margalef López, a Spanish ecologist, introduced a biodiversity measure in 1958 (*Margalef, 1973*) that emphasizes species richness. Margalef's diversity index is particularly suitable for small sample sizes and is formulated using the following formula.

$$R = (S-1)/lnN. \tag{4}$$

## Menhinick's index

Menhinick's diversity index, formulated by *Menhinick (1964)*, is based on species richness. It is calculated using the formula provided below.

Menhinick's Index

$$M = S/\sqrt{N}. \tag{5}$$

## McIntosh diversity index

Robert P. McIntosh introduced a diversity measure in 1967 (*McIntosh, 1967*), which considers both species richness and evenness. McIntosh's diversity measure utilizes the Euclidean distance measure and is calculated using the mathematical formula provided below

$$D_m = (N-U)/(N-\sqrt{N}). \tag{6}$$
## Berger-Parker's index

Berger-Parker's diversity index, introduced by *Berger & Parker (1970)*, estimates the proportion of relative species abundance. It is calculated using the formula shown below

$$d = N_{max}/N \ or \ d = 1 - N_{max}/N. \tag{7}$$

In the formula, $N_{max}$ represents the number of individuals in the most abundant species, and N represents the total number of individuals in the sample. This diversity index focuses on species richness when assessing biodiversity. *Caruso et al. (2006)* experimented and found that this index is effective in monitoring biodiversity of distributed soils.

In addition to these measures, several other measures also exist such as Hurlbert's index (*Hurlbert, 1971*), Fager's Indices (*Fager, 1972*), Hill numbers (*Hill, 1973*), Keefe and Bergersen's Index (*Keefe & Bergersen, 1977*). Rao's Diversity and dissimilarity coefficients (*Rao, 1982*).

While diversity indices are crucial tools in ecological research, each comes with its own set of disadvantages. Simpson's diversity index, though simple to calculate and interpret, tends to ignore rare species and operates within a limited range, which can be less intuitive. Shannon's diversity index, despite being sensitive to both rare and common species and widely used, suffers from complex calculation and sensitivity to sample size, necessitating larger datasets for accuracy. Brillouin's index, ideal for small, complete samples, is complex to compute and less commonly used, hindering cross-study comparisons. Margalef's richness index, although easy to compute and effective with large samples, fails to account for species evenness and is highly sensitive to sample size. Menhinick's index, similarly simple, also ignores evenness and is less commonly utilized, impacting comparative analyses. The McIntosh index, which considers both richness and evenness, is more robust against sample size variations but is complex to calculate and less intuitive. Berger-Parker's index, focusing on dominance, is straightforward and useful for identifying dominant species but overlooks rare species and may mislead interpretations of diversity. Hill's diversity index, despite its comprehensive approach, involves mathematical complexity, sensitivity to sample size, and can be difficult to interpret and apply correctly, particularly for non-specialists. Pielou's Equitability, while normalizing evenness, inherits the Shannon index's limitations and assumes equal sampling effort, making it contextually specific and occasionally challenging to interpret. Beta diversity indices, which are invaluable for understanding species composition variations across habitats, come with several disadvantages. One major drawback is their sensitivity to sample size and sampling effort, which can lead to biased or inconsistent results. Many indices, such as the Bray-Curtis Dissimilarity and Morisita-Horn index, require detailed species abundance data, making them resource-intensive and complex to calculate. Indices such as Jaccard and Sørensen only consider species presence or absence, ignoring abundance, which limits their informativeness in communities where abundance data are critical. Additionally, these indices may be less sensitive to rare species, potentially overlooking important ecological nuances. Hence, these combined limitations underscore the importance of developing a new mathematical model.

## METHODOLOGY

In the process of developing proposed measure, we followed the methodology depicted in the Fig. 2. It involves the following steps

**Step 1:** In this step, several scenarios closely resembling to the natural settings were considered.

**Step 2:** For each of the scenario's identified in Step 1 , we explored all of its associated sub cases in this step.

**Step 3:** In this step, we generate synthetic data tailored to the scenarios and use cases that closely resemble natural settings.

**Step 4**: On the synthetic data, we applied Simpson and Shannon measures to assess their effectiveness.

**Step 5**: Subsequent to the application of traditional measures on the synthetic data, we analysed particular scenarios in which the suitability of traditional measures is inadequate.

**Step 6**: To address the inadequacy of the traditional measures, we introduce our proposed measure.

**Step 7**: Here, we evaluate the effectiveness of the proposed measure on the same dataset.

**Step 8**: Examine the effectiveness of the proposed measure over traditional measures.

## EXPLORING REAL-TIME USE CASE SCENARIOS-SYNTHETIC DATA MODELLING

### Scenario's description

In the process of evaluation of biodiversity, we initially identified a few scenario's which closely resemble the real-time situations and identified sub cases that encompass changes in species populations over specific time periods.

**Scenarios:**
1. Species may disappear completely.
2. Species count may increase/ decrease significantly.
3. New species may be added.
4. No change in species.

### Scenario's and associated subcases

Subsequently, we considered all possible sub-cases within each of these scenario's. These sub-cases encompassed various events, including sudden increase, decrease or disappearance in species populations at particular moments.We have considered species namely S1, S2, S3, S4 and time periods T1 & T2.

**Scenario 1**: Species may disappear completely.

### Subcases

1. The species (S3) present in Time T1 vanished by the time T2 arrived.
2. The species (S3) that was highly abundant in Time T1 vanished entirely by Time T2.
3. Every species present in Time T1 vanished by the time T2 arrived.
4. Certain species that appeared in T1 completely disappeared in T2, while all other species exhibited a significant increase in population size.

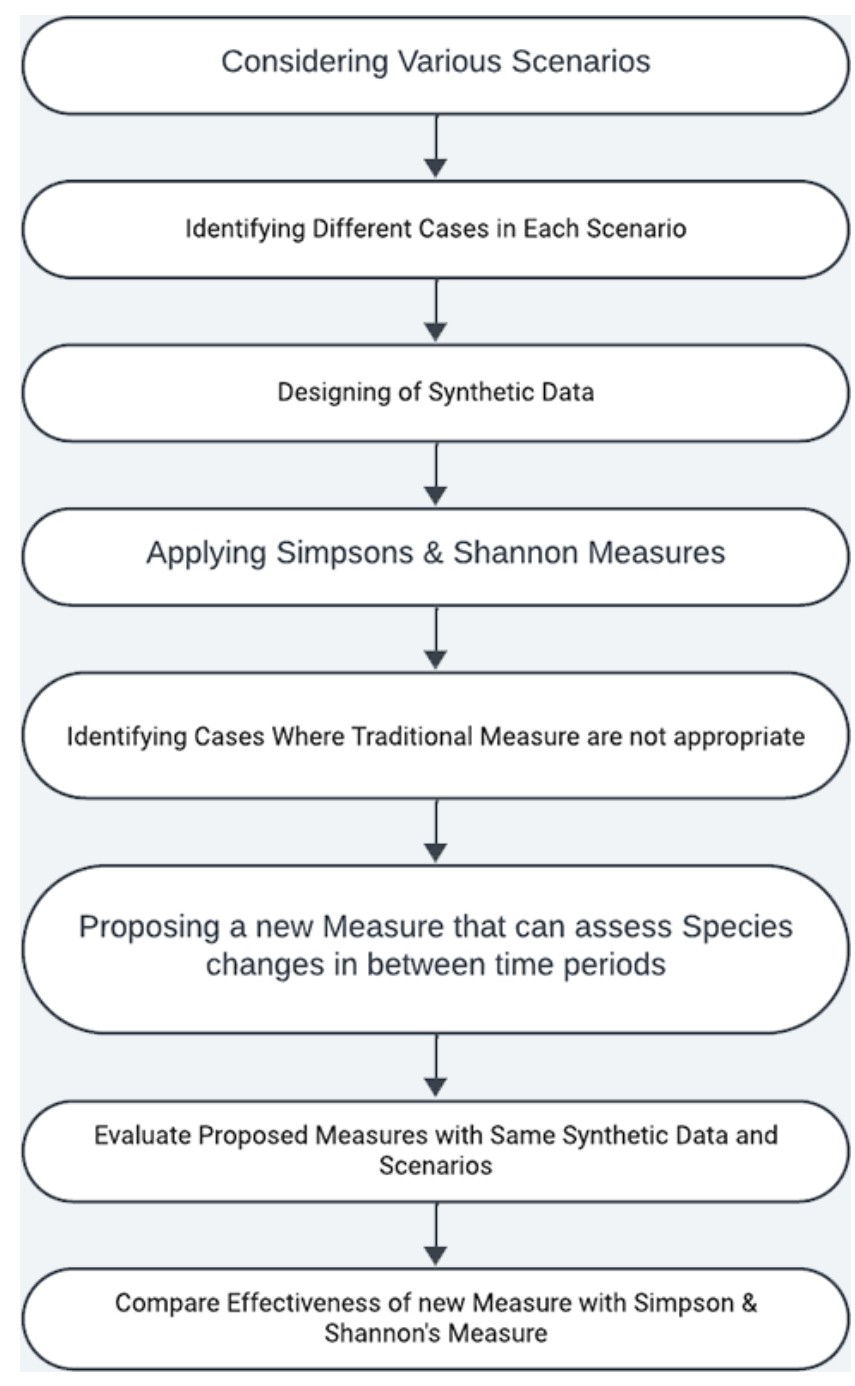

**Figure 2  Process of evaluating biodiversity.**

**Scenario 2:** Species count may increase/ decrease significantly.

### Subcases
1. All species are high in number in both time periods T1 and T2.
2. All species are less in number in both time periods T1 and T2.

3. Species S2, which was abundant in time period T1, experienced a notable decrease in abundance during time period T2.
4. Species S2, initially scarce in number during time period T1, exhibited a substantial increase in abundance during time period T2.
5. The species that were abundant during time period T1 experienced a notable decrease in population size during time period T2.
6. The species that were initially scarce in number during T1 exhibited a substantial increase in population size during T2.
7. Certain species present in T1 underwent a proportional increase in abundance by the time T2 occurred.
8. Certain species present in T1 experienced a proportional decrease in abundance by the time T2 occurred.
9. Every species present in T1 experienced a proportional increase in abundance by the time T2 occurred.
10. Every species present in T1 underwent a proportional decrease in abundance by the time T2 arrived.

   **Scenario 3:** New species may be added.

### Subcases

1. During T2, new species that were not present in T1 have been introduced.
2. At T1, no species were present, yet they emerged in notably large numbers by T2.
3. Several species absent in T1 have emerged in significant numbers during T2.

   **Scenario 4:** No change in species.

### Subcases

1. No alterations in species composition occurred during both time periods T1 and T2.

### Crafting synthetic data

Table 1 depicts the generated synthetic data tailored to scenario's and use cases closely resembling natural settings (*Díaz et al., 2020*). In generating this synthetic data, we have considered different species namely S1, S2, S3, S4 and their associated counts for two different time periods T1 & T2.

## UNVEILING MEASUREMENT GAPS: SIMPSON & SHANNON MEASURES

Simpson and Shannon are considered to be most widely used measures to assess biodiversity. Hence, we have chosen these two measures to apply on our synthetic data to identify their strengths and weaknesses. Upon applying the Simpson and Shannon measures on the data,the results were tabulated in Table 2.

From the results obtained in Table 2, we have identified specific scenarios where the applicability of traditional measures are falling short.

   **Cases where Shannon's measure is failing**

1. Every species present in time T1 vanished by the time T2 arrived.

**Table 1  Synthetic data to evaluate biodiversity.**

| Scenario | S1 | | S2 | | S3 | | S4 | |
|---|---|---|---|---|---|---|---|---|
| | T1 | T2 | T1 | T2 | T1 | T2 | T1 | T2 |
| No alterations in species composition occurred during both time periods T1 and T2. | 10 | 10 | 19 | 19 | 23 | 23 | 22 | 22 |
| The abundance of all species remains consistently high across both time periods T1 and T2. | 930 | 940 | 950 | 1,000 | 1,300 | 1,260 | 850 | 750 |
| The population of all species exhibits a decrease during both time periods T1 and T2. | 1 | 1 | 3 | 2 | 2 | 2 | 1 | 4 |
| Species S2, which was abundant in time period T1, experienced a notable decrease in abundance during time period T2. | 45 | 43 | 450 | 4 | 46 | 39 | 54 | 52 |
| Species S2, initially scarce in number during time period T1, exhibited a substantial increase in abundance during time period T2. | 20 | 180 | 50 | 2,100 | 100 | 400 | 150 | 850 |
| The species that were abundant during time period T1 experienced a notable decrease in population size during time period T2. | 180 | 1 | 200 | 2 | 170 | 1 | 230 | 2 |
| The species that were initially scarce in number during T1 exhibited a substantial increase in population size during T2. | 4 | 880 | 3 | 760 | 3 | 990 | 3 | 890 |
| Certain species present in T1 underwent a proportional increase in abundance by the time T2 occurred. | 10 | 8 | 23 | 230 | 45 | 450 | 22 | 26 |
| Certain species present in T1 experienced a proportional decrease in abundance by the time T2 occurred. | 100 | 10 | 200 | 20 | 210 | 209 | 220 | 215 |
| Every species present in T1 experienced a proportional increase in abundance by the time T2 occurred. | 10 | 100 | 18 | 180 | 21 | 210 | 22 | 220 |
| Every species present in T1 underwent a proportional decrease in abundance by the time T2 arrived. | 100 | 10 | 120 | 12 | 80 | 8 | 90 | 9 |
| The species (S3) present in Time T1 vanished by the Time T2 arrived. | 10 | 10 | 9 | 7 | 32 | 0 | 22 | 23 |
| The species (S3) that was highly abundant in Time T1 vanished entirely by Time T2. | 10 | 10 | 22 | 22 | 430 | 0 | 25 | 25 |
| Every species present in Time T1 vanished by the Time T2 arrived. | 180 | 0 | 110 | 0 | 190 | 0 | 220 | 0 |
| Certain species that appeared in T1 completely disappeared in T2, while all other species exhibited a significant increase in population size. | 25 | 0 | 32 | 0 | 45 | 2,100 | 54 | 2,200 |
| During T2, new species that were not present in T1 have been introduced. | 18 | 13 | 0 | 20 | 17 | 18 | 21 | 22 |
| At T1, no species were present, yet they emerged in notably large numbers by T2. | 0 | 1,300 | 0 | 1,100 | 0 | 2,100 | 0 | 2,200 |
| Several species absent in T1 have emerged in significant numbers during T2. | 180 | 515 | 170 | 672 | 0 | 2,100 | 173 | 472 |

**Table 2  Applying simpson and shannon formula to the synthetic data.**

| Scenario | Shannon | Simpson |
|---|---|---|
| No alterations in species composition occurred during both time periods T1 and T2. | 0 | 0 |
| The abundance of all species remains consistently high across both time periods T1 and T2. | −0.00291 | −0.00116 |
| The population of all species exhibits a decrease during both time periods T1 and T2. | −0.00401 | −0.03175 |
| Species S2, which was abundant in time period T1, experienced a notable decrease in abundance during time period T2. | 0.368672 | 0.276419 |
| Species S2, initially scarce in number during time period T1, exhibited a substantial increase in abundance during time period T2. | −0.08775 | −0.05508 |
| The species that were abundant during time period T1 experienced a notable decrease in population size during time period T2. | −0.04984 | 0.11916 |
| The species that were initially scarce in number during T1 exhibited a substantial increase in population size during T2. | 0.004144 | −0.05963 |
| Certain species present in T1 underwent a proportional increase in abundance by the time T2 occurred. | −0.43391 | −0.19487 |
| Certain species present in T1 experienced a proportional decrease in abundance by the time T2 occurred. | −0.41423 | −0.17098 |
| Every species present in T1 experienced a proportional increase in abundance by the time T2 occurred. | 0 | −0.00943 |
| Every species present in T1 underwent a proportional decrease in abundance by the time T2 arrived. | 0 | 0.017672 |
| The species (S3) present in Time T1 vanished by the Time T2 arrived. | −0.2686 | −0.10152 |
| The species (S3) that was highly abundant in Time T1 vanished entirely by Time T2. | 0.563354 | 0.423365 |
| Every species present in Time T1 vanished by the Time T2 arrived. | 0.028525 | −1.23779 |
| Certain species that appeared in T1 completely disappeared in T2, while all other species exhibited a significant increase in population size. | −0.65084 | −0.23407 |
| During T2, new species that were not present in T1 have been introduced. | 0.262873 | 0.075929 |
| At T1, no species were present, yet they emerged in notably large numbers by T2. | 1.344109 | 0.729447 |
| Several species absent in T1 have emerged in significant numbers during T2 | 0.055617 | −0.04618 |

### Cases where Simpson's measure is failing

1. The species that were abundant during time period T1 experienced a notable decrease in population size during time period T2.
2. The species that were initially scarce in number during T1 exhibited a substantial increase in population size during T2.

Identification of cases where both Simpsons and Shannon's measure are failing and the inference behind their failure is shown in the Table 3.

## EXPLORING PROPOSED MEASURE

We can understand that several flaws exists in Simpson's and Shannon's measures, which are discussed below.

1. Both Shannon and Simpson's measures are sensitive to sample size making it challenging to compare diversity across different sites with varying sample sizes.

2. Shannon measure is more inclined to the abundance of species, which may have little impact on the index, potentially leading to an underestimation of the importance of rare or unique species in a community.

3. Shannon does not distinguish between dominant and rare species, which can be a drawback in situations where a few highly dominant species might have a disproportionate impact on the ecosystem compared to a larger number of rare species.

4. Shannon's diversity index may not be very sensitive to changes in species richness, especially when comparing communities with similar evenness but different numbers of species. This can make it challenging to detect changes in biodiversity if species richness is the primary concern.

5. Simpson's index is highly influenced by the dominance of one or a few species in a community resulting in an underestimation of diversity when there is dominance of a few species.

6. Simpson's index focuses on the dominance of one or a few species and does not explicitly account for the evenness of the distribution of individuals among different species.

7. Simpson's index provides a combined measure of richness and evenness, making it difficult to differentiate the impact of changes in species richness from changes in evenness.

To address these gaps, we proposed a mathematical model which is described in the subsequent sections.

## Design of proposed mathematical measure

These limitations emphasize the need for robust measures to assess biodiversity. In the computation of biodiversity, it is very much essential to calculate the evenness and richness separately unlike Simpson and Shannon.

We formulated our mathematical model which is robust enough in computing the impact of biodiversity when there is a transition of species across different time periods without inclining to any factors such as species dominance and sample size. Further this mathematical model computes evenness and richness separately so that it will be very effective in differentiating the changes of species.

In line with these views, we have proposed a new mathematical model for computing biodiversity as follows. Firstly, we have calculated the deviation of each individual species in the time T1. Based on these values we have calculated the average deviation of all species. To derive the evenness, we subtracted the evenness whole value with the average deviation. The generic evenness formula is as follows:

$$Evenness = (N_S - \sum_{i=1}^{N_S} \sqrt{(1/N_S - a_i/N)^2})/N_S \tag{8}$$

where,

$a_i$ = No. of species of 'a' in time period $T_i$

$N_S$ = No. of different species

**Table 3  Cases where Simpson and Shannon are failing.**

| Cases | Inference of Simpsons | Inference of Shannon |
|---|---|---|
| • All species are high in number in both time periods T1 and T2.<br>• Species S2, which was abundant in time period T1, experienced a notable decrease in abundance during time period T2.<br>• Species S2, initially scarce in number during time period T1, exhibited a substantial increase in abundance during time period T2.<br>• Certain species present in T1 underwent a proportional increase in abundance by the time T2 occurred.<br>• Every species present in T1 experienced a proportional increase in abundance by the time T2 occurred.<br>• Every species present in T1 underwent a proportional decrease in abundance by the time T2 arrived.<br>• The species (S3) present in Time T1 vanished by the Time T2 arrived.<br>• Certain species that appeared in T1 completely disappeared in T2, while all other species exhibited a significant increase in population size<br>• Several species absent in T1 have emerged in significant numbers during T2. | • This measure demonstrates effectiveness in scenarios where there is a consistent absence of species change between periods.<br>• In instances where species disappear between time periods, this measure can only effectively identify cases where a small number of species vanish. However, its sensitivity is limited, as it fails to discern instances where a large number of species disappear.<br>• Regarding the addition of new species, this measure also lacks sensitivity. It does not accurately depict situations where a substantial number of species have been introduced.<br>• In certain scenarios involving changes in species composition between periods, the Simpson's index difference values may be inadequate for drawing conclusions and fail to identify these changes effectively. | • Shannon's index becomes unreliable when there is a loss of species data (i.e., when species count equals zero) included in the analysis.<br>• Even after excluding species with zero values, the method remains insensitive to detecting changes in certain instances.<br>• The measure lacks sensitivity in accurately indicating a substantial number of additions of new species.<br>• The measure fails to recognize instances where certain species are lost while others experience an increase between time periods.<br>• The measure indicates no change in biodiversity (stability) despite a proportional increase in species count between time periods. |

**Table 4  Interpretation of values.**

| Value | Interpretation |
|---|---|
| 0.5 | No alteration in biodiversity observed between the two time periods. |
| $> 0.5$ | Indicates a raise in biodiversity between two time periods. |
| $< 0.5$ | Indicate decrease in biodiversity between two time periods. |

$N$ = Total count of different species in T1

Now, we calculate the biodiversity value using the formula shown in Eq. (9) and by utilizing the above computed evenness value in the Eq. (8).

$$Biodiversity = \frac{\sum_{i=1}^{N_S} \frac{a_i}{T_i} * Evenness}{N_S} \qquad (9)$$

where,

$a_i$ = No. of species of 'a' in time period $T_i$

$T_i$ = Total count of the $i$th individual species

Repeat the above process for different time periods. Through this repetition process for different time periods (*e.g.*,: T1, T2), we obtained two different biodiversity values (biodiversity at time T1 & T2). The difference of these two values indicates the change in biodiversity.

The disparity in values can be understood as provided in Table 4.

To understand which individual species have made an impact on biodiversity can be calculated by using the formula given in Eq. (10).

$$Biodiversity(k) = \frac{\frac{a_k}{T_i} * Evenness + \sum_{\substack{i=1 \\ i \neq k}}^{N_S} \frac{a_i}{T_i} * Evenness}{N_S}. \qquad (10)$$

Further the effectiveness of the proposed mathematical model will be evaluated in the next section by considering the same synthetic data which we considered for evaluation in Table 1.

## COMPARATIVE ANALYSIS: PROPOSED MEASURE VERSUS TRADITIONAL METHODS

Upon utilising the same synthetic data as in Table 1, biodiversity is computed using the proposed mathematical measure. The computed values obtained by applying proposed measure are tabulated in column 2 of Table 5.

From the results obtained through Table 5, it can be inferred that Shannon and Simpson measures are identified to be insensitive to the species changes. However, when evaluated and compared the proposed measure with the existing traditional methods, proposed measure has shown nice degree of sensitivity in species changes with regard to the following parameters.

### Insensitive to the sample changes

One of the significant drawbacks of traditional methods lies in the insensitivity of both Shannon and Simpson's measures towards sample size (*Soetaert & Heip, 1990*), posing a challenge in accurately assessing biodiversity. In order to be sensitive, any measure should be able to detect minor or major changes in the species. When we tried different species combinations and experimented to evaluate the sensitivity, proposed measure was effective in the following use cases.

- A species that was abundant in time period T1 experienced a notable decrease in population size during time period T2.
- A species that was initially scarce in number during time T1 underwent a significant increase in population size during time T2.
- Certain species present in T1 exhibited a proportional increase in abundance during T2.
- All species present in T1 underwent a proportional decrease in abundance during Time T2.
- All species that were abundant during time period T1 exhibited a notable decrease in population size during time period T2.

The proposed measure can detect species changes in all circumstance. For instance, it is sensitive to sudden increase or decrease of single species in any of the time periods. Also proposed measure effectively recognises even the increase or decrease of multiple species in the given time periods.

### Inclination to the dominance of few species

Another significant drawback of the traditional method is inclining to the more dominant or abundant species which is resulting in an underestimation of diversity. To eliminate dominance bias, the proposed measure aims to be free from any influence of dominant species. To validate scenarios involving dominance bias, we have examined various combinations of dominance conditions. These included use cases such as significant
**Table 5  Values obtained through proposed measure on same synthetic data.**

| Scenarios | Proposed measure | Shannon | Simpson |
|---|---|---|---|
| No alterations in species composition occurred during both time periods T1 and T2. | 0.485 | 0 | 0 |
| The abundance of all species remains consistently high across both time periods T1 and T2. | 0.4777 | −0.00291 | −0.00116 |
| The population of all species exhibits a decrease during both time periods T1 and T2. | 0.5158 | −0.00401 | −0.03175 |
| Species S2, which was abundant in time period T1, experienced a notable decrease in abundance during time period T2. | 0.3273 | 0.368672 | 0.276419 |
| Species S2, initially scarce in number during time period T1, exhibited a substantial increase in abundance during time period T2. | 0.7735 | −1.20402 | −0.70181 |
| The species that were abundant during time period T1 experienced a notable decrease in population size during time period T2. | 0.007 | −0.04984 | 0.11916 |
| The species that were initially scarce in number during T1 exhibited a substantial increase in population size during T2. | 0.9793 | 0.004144 | −0.05963 |
| Certain species present in T1 underwent a proportional increase in abundance by the time T2 occurred. | 0.5843 | −0.43391 | −0.19487 |
| Certain species present in T1 experienced a proportional decrease in abundance by the time T2 occurred. | 0.2467 | −0.41423 | −0.17098 |
| Every species present in T1 experienced a proportional increase in abundance by the time T2 occurred. | 0.8842 | 0 | −0.00943 |
| Every species present in T1 underwent a proportional decrease in abundance by the time T2 arrive . | 0.088 | 0 | 0.017672 |
| The species (S3) present in Time T1 vanished by the Time T2 arrived. | 0.3032 | −0.2686 | −0.10152 |
| The species (S3) that was highly abundant in Time T1 vanished entirely by Time T2. | 0.3211 | 0.563354 | 0.423365 |
| Every species present in Time T1 vanished by the Time T2 arrived. | 0.006 | 0.028525 | −1.23779 |
| Certain species that appeared in T1 completely disappeared in T2, while all other species exhibited a significant increase in population size. | 0.3971 | −0.65084 | −0.23407 |
| During T2, new species that were not present in T1 have been introduced. | 0.5992 | 0.262873 | 0.075929 |
| At T1, no species were present, yet they emerged in notably large numbers by T2. | 0.9424 | 1.344109 | 0.729447 |
| Several species absent in T1 have emerged in significant numbers during T2. | 0.7145 | −0.3968 | −0.33093 |

increase in all species populations between time periods, proportional increase across all species, non-proportional increase across all species, substantial increase in particular species while others remain nominal, few species disappearances between time periods while all other species exhibiting substantial population growth, and the significant addition of new species in any of the time periods. When tested with all these use cases proposed

measure traded-off well in the following use cases when compared to the traditional methods.

- All species that were initially scarce in number during T1 experienced a substantial increase in population size during T2.
- Every species present in T1 experienced a proportional increase in abundance during Time T2.
- A species that was initially scarce in number during T1 underwent a significant increase in population size during T2.
- Certain species that appeared in T1 completely disappeared by T2, while all other species exhibited significantly large populations.

## Ignoring rare species identity

Ignoring the presence of rare species is another flaw that exists in traditional methods. It is very important to consider the presence of rare species as it plays crucial role in maintaining the ecosystem diversity. However, traditional methods ignored the identity of rare species as they focused on the abundance of species. Hence these methods present challenges in detecting changes in biodiversity. The proposed measure was modelled to overcome this limitation of underestimation of rare species.

The proposed measure is examined by taking various scenarios into consideration such as addition of new species in any of the time periods, complete disappearance of any species in particular time periods, changes in the species abundance such as sudden increase in all species exists in time period T1 to T2 and sudden disappearance in all species from one time period to other. When we tried different species combinations and experimented to evaluate the identity of the rare species proposed measure was effective in the following use cases.

- Certain species absent in T1 have emerged in large numbers during T2.
- Any species that was highly abundant in Time T1 vanished entirely by Time T2.
- Every species present in Time T1 vanished by the time T2 arrived.

The proposed measure for identifying rare species has demonstrated effectiveness across various scenarios, including the addition of new species, complete disappearance of certain species, and changes in species abundance over time. Through rigorous experimentation and evaluation of different species combinations, we observed compelling evidence supporting the efficacy of the proposed measure in multiple use cases to identify the rare species.

The outcomes across all scenarios consistently demonstrate the effectiveness of the proposed measure in rare species identification. Whether dealing with dominant species, varying sample sizes, or the identity of rare cases, the measure proves reliable and versatile. Its ability to accurately identify new species introductions, sudden disappearances, and broad-scale changes in species abundance underscores its importance in biodiversity conservation efforts. By providing a robust framework that transcends ecological complexities, the proposed measure significantly contributes to advancing conservation strategies, ensuring the preservation of precious biodiversity for generations to come.

## DISCUSSION

From the results we could infer that our proposed measure for biodiversity assessment demonstrates significant advantages in detecting species changes across various conditions. The measure exhibits high sensitivity to sudden increases or decreases in single species over time and effectively identifies changes involving multiple species within the same time frame. To address dominance bias, we tested the measure against a variety of dominance scenarios, including:

- Significant overall population increases
- Proportional and non-proportional species increases
- Substantial growth in particular species
- Species disappearances
- Introduction of new species

Through these rigorous validations, our proposed measure consistently outperformed traditional methods. Additionally, it accounts for the identity of rare species by evaluating scenarios such as:

- Addition or disappearance of species
- Sudden increases in all species from T1 to T2
- Sudden disappearance of all species between periods

The proposed measure ensures rare species are accounted for, proving effective in various scenarios such as the emergence of previously absent species, the complete disappearance of highly abundant species, and the total disappearance of all species over time. The graph showing the performance of the proposed measure over traditional methods is illustrated in Fig. 3.

Our proposed measure for biodiversity assessment offers significant advantages by being highly sensitive to species changes under various circumstances. It can detect sudden increases or decreases in single species across time periods, as well as effectively recognize multiple species changes within the same time frame. To address dominance bias, our measure eliminates the influence of dominant species by testing a variety of dominance scenarios, including significant overall population increases, proportional and non-proportional species increases, substantial growth in particular species, species disappearances, and the introduction of new species. Through these rigorous validations, our proposed measure consistently outperformed traditional methods. These contributions demonstrate the robustness and reliability of our measure in accurately assessing biodiversity changes while mitigating common biases and limitations.

## CONCLUSION

Our research underscores the potential for incorporating innovative features into established biodiversity conservation frameworks. The introduction of our proposed mathematical model marks a notable stride forward in biodiversity assessment methodologies. This model not only enhances efficacy and resilience compared to current methods but also provides a more robust approach to comprehending and safeguarding biodiversity within complex and dynamic ecosystems. By evaluating biodiversity over

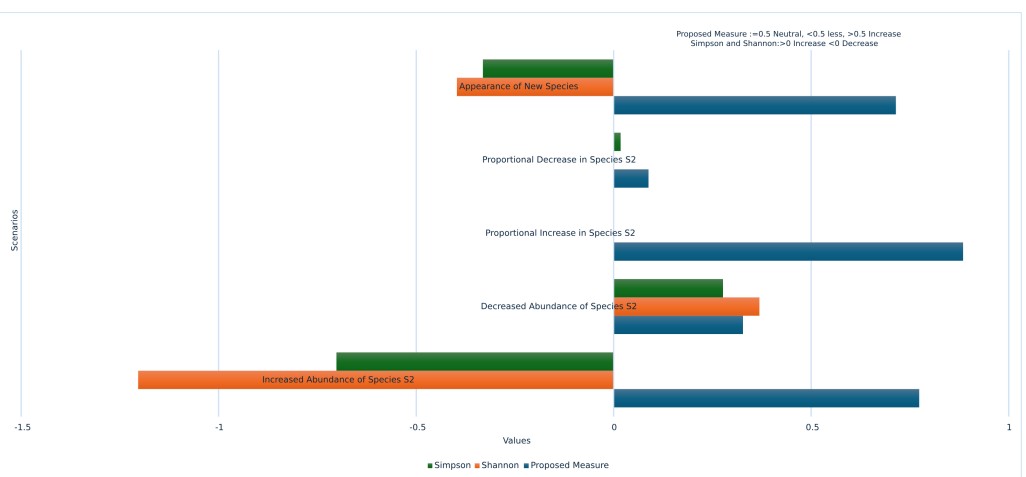

**Figure 3** Performance of proposed measure *vs* traditional measures.

distinct time periods, it offers valuable insights into its fluctuating nature. Given that biodiversity assessment is an ongoing process necessitating continual refinement and adaptation to evolving environmental conditions and scientific insights, future endeavors in this field are likely to prioritize areas such as the assessment of functional diversity, integration of genomic data, advanced remote sensing and spatial analysis techniques, dynamic predictive modelling, *etc*. However, these endeavours face several limitations, including challenges related to data availability and quality, scalability of models, complexity of ecosystem interactions. Addressing these limitations and advancing future research directions will significantly contribute to a better understanding of biodiversity and ecosystem functioning, ultimately facilitating more effective conservation and management strategies.

## Funding
The authors received no funding for this work.

## Competing Interests
The authors declare there are no competing interests.

## Author Contributions
- Manjula Josephine Bollarapu conceived and designed the experiments, performed the experiments, analyzed the data, prepared figures and/or tables, authored or reviewed drafts of the article, and approved the final draft.
- Swarna Kuchibhotla conceived and designed the experiments, analyzed the data, prepared figures and/or tables, and approved the final draft.
- Ramarao Kvsn conceived and designed the experiments, authored or reviewed drafts of the article, and approved the final draft.

- Harshita Patel analyzed the data, authored or reviewed drafts of the article, and approved the final draft.

## Data Availability

The raw data and code are available as Supplemental Files.

## Supplemental Information

Supplemental information for this article can be found online at http://dx.doi.org/10.7717/peerj.17924#supplemental-information.

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
