# Peer review of "Dynamic perspectives on biodiversity quantification: beyond conventional metrics"

_PeerJ, doi:10.7717/peerj.17924_

## Round 0.1 · original submission · Major Revisions

Dear Authors,

I have now received the necessary reviews to make a decision on the manuscript. After reading the text and the reviewers' comments, I have decided to major revisions. The material presented is interesting and relevant to studies in ecology and will add another important tool for practical work in this science. However, as the text stands, some substantial changes are necessary. I would like to draw attention to the comments of reviewer number 1, as they point out that there is a greater need for theoretical contextualization, as well as a more concise presentation of the limitations of current indices. In addition, the reviewer points out that there is a need for better presentation of the results, as well as bringing them into line with current ecological theory.

Sincerely,

·

Basic reporting

This paper reports on interesting work on the development of an index that can better describe biodiversity and its variation through time. While the goals of the manuscript and the approach are certainly commendable, there is a couple of issues that arisen during my reading that seems critical, including in data analysis and reproducibility and theoretical background.
On the theoretical background end, the authors present a surplus of indices they are not actually approaching in the text and fail to explore the literature critiquing diversity indices and pathways for better measuring biodiversity. For example, they emphasize that alpha diversity indices fail in showcase biodiversity dynamics, but fail to showcase that it is not the objective of alpha diversity indices to showcase variation between sites or periods, and that temporal dynamics are actually considerably done using beta diversity indices (and they use the same pathway of calculating a difference between T1 and T2 to showcase it). In addition, they recognize the main issues with most diversity indices, but fail to showcase the pathways already in place for solving such issues, such as the Hills numbers (despite citing its existence), or simply using richness and Pielou’s equability separately. See below for my comments on the methodological choices.
Presentation-wise, a better use of graphs could be done to showcase how the variation in one or a couple of species can affect a given index. For example, a graph with the Proportional abundance of S1 from 5% to 80%, and the standardized value of the index on the Y-axis could showcase the differences between the chosen indices and the proposed by the authors. Finally, the text should be language edited to ensure clarity for a broad audience. Throughout out the text, there are instances where wording is wrong (declination of biodiversity, “of these scenario’s”) or confusing (“minor or major changes in the species”… abundances?).

Experimental design

Material and methods are not clear enough to secure reproducibility. The authors do not disclose the program(s) used for data analysis and the entire protocol for index calculations and comparisons. In addition, the created scenarios are poorly standardized and there is no theoretical background supporting the decision on the used numbers, thus inducing misinterpretations. For example, in the second scenario, the abundance of species remains high, but some species exhibit increased abundance and other decreased abundance. Their changes in relative abundance are also not standardized (what is a notable decrease? And a substantial increase?) and the raw values of biodiversity and biodiversity change would probably play a role in biodiversity calculation, but it is unclear what is such a role (i.e., what does it mean high abundance? 100? 1000? 10000?). Also, a couple of the scenarios seems to be misleading. For example, the 5th scenario showcases a increase in the Biodiversity index proposed by the authors but a decrease in Shannon’s and Simpson’s diversity from T1 to T2. When we look at the input data, we notice that every species had their abundances increased at least four times from T1 to T2. Looking at the data, I also notice that some of the results might be strongly driven by total abundance in a way that increases in the abundance of a given species will necessarily increase biodiversity. Of course, this might be a desired property for an index, while it is not usually desired in common biodiversity metrics because of sampling efficiency issues; so authors should also explore this.
In this sense, I strongly recommend authors to reduce the number of explored scenarios and better standardize how they generate community composition variation; otherwise, the robustness of the assessment is compromised. For example, authors might establish that, for standardization, they would work with ecological communities varying from 4 individuals to 1000 individuals. Then, calculate the scenarios based on how the addition of individuals in the initial community of 4 individuals (one for each species) changes the analyzed indices. A loop could be created for adding one individual at a time and calculating the indices, while maintaining the proper configuration of the other species.

Validity of the findings

The findings are not robust given the choices in the experimental design and it is unclear what the proposed index is actually measuring.

Reviewer 2 ·

Basic reporting

- Contributions should be highlighted.
- It is advised to discuss the benefits and drawbacks of the literature.
- Please check equation 2 and 3.
- Discussion should be added, regarding findings
- In Conclusion, add future research directions with study limitations.
- The article needs to be a review of grammatical errors.

Experimental design

How Synthetic data is generated. Please discuss.

Validity of the findings

What are the findings.
Describe experimental results.

---

## Round 0.2 · accepted · Accept

The authors have addressed all of the reviewer's comments.

Reviewer 2 ·

Basic reporting

Authors has addressed my concerns. Paper my be accepted.

Experimental design

NA

Validity of the findings

NA

Additional comments

NA